# Inappropriate Evaluation of Effect Modifications Based on Categorical Outcomes: A Systematic Review of Randomized Controlled Trials

**DOI:** 10.3390/ijerph192215262

**Published:** 2022-11-18

**Authors:** Akihiro Shiroshita, Norio Yamamoto, Natsumi Saka, Motohiro Okumura, Hiroshi Shiba, Yuki Kataoka

**Affiliations:** 1Department of Respiratory Medicine, Ichinomiyanishi Hospital, Ichinomiya 494-0001, Japan; 2Division of Epidemiology, Department of Medicine, Vanderbilt University School of Medicine, Nashville, TN 37203, USA; 3Scientific Research WorkS Peer Support Group (SRWS-PSG), Osaka 541-0043, Japan; 4Department of Epidemiology, Graduate School of Medicine, Dentistry and Pharmaceutical Sciences, Okayama University, Okayama 700-8558, Japan; 5Department of Health Research Methods, Evidence & Impact, McMaster University, Hamilton, ON L8S 4K1, Canada; 6Department of Orthopaedic Surgery, Teikyo University School of Medicine, Tokyo 173-8606, Japan; 7Department of Neurology, Jikei University School of Medicine, Tokyo 105-8471, Japan; 8Department of Internal Medicine, Suwa Central Hospital, Chino 391-8503, Japan; 9Department of Internal Medicine, Kyoto Min-Iren Asukai Hospital, Kyoto 606-8226, Japan; 10Section of Clinical Epidemiology, Department of Community Medicine, Kyoto University Graduate School of Medicine, Kyoto 606-8501, Japan; 11Department of Healthcare Epidemiology, Kyoto University Graduate School of Medicine/Public Health, Kyoto 606-8501, Japan

**Keywords:** effect modification, meta-epidemiological study, nonlinear regression, randomized controlled trial, relative scale outcome

## Abstract

Our meta-epidemiological study aimed to describe the prevalence of reporting effect modification only on relative scale outcomes and inappropriate interpretations of the coefficient of interaction terms in nonlinear models on categorical outcomes. Our study targeted articles published in the top 10 high-impact-factor journals between 1 January and 31 December 2021. We included two-arm, parallel-group, interventional superiority randomized controlled trials to evaluate the effects of modifications on categorical outcomes. The primary outcomes were the prevalence of reporting effect modifications only on relative scale outcomes and that of inappropriately interpreting the coefficient of interaction terms in nonlinear models on categorical outcomes. We included 52 articles, of which 41 (79%) used nonlinear regression to evaluate effect modifications. At least 45/52 articles (87%) reported effect modifications based only on relative scale outcomes, and at least 39/41 (95%) articles inappropriately interpreted the coefficient of interaction terms merely as indices of effect modifications. The quality of the evaluations of effect modifications in nonlinear models on categorical outcomes was relatively low, even in randomized controlled trials published in medical journals with high impact factors. Researchers should report effect modifications of both absolute and relative scale outcomes and avoid interpreting the coefficient of interaction terms in nonlinear regression analyses.

## 1. Introduction

In clinical medicine, treatment effects can vary among patients; thus, precision or personalized medicine based on patient characteristics has been advocated [1]. Therefore, the evaluation of effect modifications (i.e., different treatment effects dependent on other variables) has been used as a means to identify specific patient subgroups that may respond to treatment [2]. However, when evaluating effect modifications, researchers should be cautious about the outcome scales used, as the directions of the results may not necessarily match. Two types of scales are typically used, namely, absolute scale (i.e., absolute difference) and relative scale (i.e., relative difference). As an effect modification on an absolute scale implies a net benefit of different treatment effects within a subgroup, evaluating effect modifications based not only on a relative scale but also on an absolute scale would be better for identifying specific patient subgroups [2,3,4].

For instance, let us consider a situation in which we evaluate an effect modification between a treatment and an independent variable, X, based on the treatment success rate (Appendix A). The absolute differences in treatment success rates between the treatment and non-treatment groups are 16% − 8% = 8% in the X+ group and 8% − 4% = 4% in the X− group. These results suggest the presence of an effect modification. In contrast, the relative differences are 16%/8% = 2 times in the X+ group and 8%/4% = 2 times in the X− group, which suggests no effect modification. If researchers evaluate effect modifications based only on a relative scale and not on an absolute scale, they cannot correctly identify subgroups.

The same logic applies to regression. In linear regression analyses, the coefficients indicate an absolute scale, while in nonlinear regression analyses, these indicate a relative scale. For example, a coefficient of a logistic regression indicates the change of the outcome on a log-odds ratio scale relative to a one-unit change in the corresponding covariate. It is caused by the link function (e.g., logit for logistic regression and log for a Poisson regression). Furthermore, because of the link function, nonlinear regression analyses show an inherent interaction, i.e., treatment effects are constant on a relative scale (e.g., the log-odds scale for logistic regression), whereas the probability of the outcomes changes depending on other variables without interaction terms [5,6]. For instance, as illustrated in Appendix A, a logit curve moves upwards when the intercept only increases. The change in probability is small at both extreme dependent values while it is large at the middle (i.e., compression). This phenomenon has an inherently interactive nature because the difference between the two logistic curves is contingent on the value of an independent variable. It explains the interactions between one independent variable and another independent variable without any interaction terms. In other words, just including a different independent variable without the interaction term changes the residual variation of the underlying model and moves a logit curve similarly. This inherent interactive nature was summarized by Mize et al. [7]. Thus, the coefficients of the interaction terms do not necessarily indicate any effect modification. The American Sociological Association does not recommend using the coefficient of interaction terms to evaluate effect modifications in nonlinear regression analyses of categorical outcomes [8].

In economics and sociology, inappropriate evaluations of effect modifications based on categorical outcomes have been demonstrated [5,8]. Most previous studies have not correctly interpreted the coefficients of the interaction terms in nonlinear regression analyses [5,7]. Moreover, in clinical medicine, researchers utilize many categorical outcomes, including hard outcomes (e.g., 30-day mortality rate and readmission) and soft outcomes (e.g., number of readmissions and exacerbations). However, to the best of our knowledge, no study has evaluated whether researchers in clinical medicine appropriately evaluate effect modifications based on categorical outcomes. In this study, we aimed to describe the prevalence of reporting effect modifications based only on relative scale outcomes and that of inappropriately interpreting the coefficient of nonlinear models on categorical outcomes. We targeted randomized controlled trials (RCTs) reported in high-impact-factor medical journals, which are expected to be of high quality.

## 2. Materials and Methods

### 2.1. Study Design

This was a meta-epidemiological study that used previously published RCTs. As it used only open data, informed consent from patients was not required, and there were no ethical concerns. The study protocol was pre-registered on an open platform (https://osf.io/snpj7/ registration date: 22 May 2022). Additionally, we have reported the study according to the guidelines for meta-epidemiological studies [9].

### 2.2. Eligibility Criteria

The inclusion criteria were full-text articles of two-arm, parallel-group, interventional superiority RCTs that conducted statistical analysis for evaluating subgroup-specific effect modifications between treatments and independent variables based on categorical outcomes. We included studies with factorial or cluster designs. In our analysis, we defined nonlinear regressions for categorical outcomes regardless of the frequentist or Bayesian framework as follows: (1) generalized linear models such as logistic, Poisson, negative binomial, ordinal, and multinomial logistic regression; (2) generalized linear mixed-effects models such as mixed-effects logistic, Poisson, negative binomial, ordinal, and multinomial logistic regression; and (3) generalized estimating equations. We did not restrict the types of outcomes, such as primary, secondary, and explanatory outcomes. The exclusion criteria were articles of other types of RCTs, such as equivalence, non-inferiority, cross-over, or more than two treatment group trials, as well as study protocols and animal studies. Moreover, when only a two-by-two table of outcomes was created, and researchers did not analyze the effect modification, we excluded the related articles. Additionally, when effect modifications were planned in the study protocols, and the results were not described in the main text or supplementary materials, the related articles were excluded. Finally, articles published in non-English languages were excluded.

### 2.3. Search Strategy

We searched for potential RCTs in high-impact-factor medical journals published between 1 January and 31 December 2021. Based on the journal impact factors from journal citation reports in 2020, the major clinical journals (a category of clinical medicine) were as follows: *The New England Journal of Medicine*, *The Lancet*, *Journal of the American Medical Association*, *The British Medical Journal*, *Annals of Internal Medicine*, *JAMA Internal Medicine*, *Lancet Oncology*, *Journal of Clinical Oncology*, *Lancet Neurology*, and *Lancet Infectious Diseases* [10]. A.S. searched for potential studies in Medline via PubMed using related search terms (Appendix A).

### 2.4. Study Selection

For screening, A.S. reviewed the titles and abstracts of the selected articles and checked whether they met the inclusion criteria. A.S. then reviewed the full text and online appendixes to determine whether the articles would be finally included. Any one of the remaining authors (N.Y., N.S., M.O., or H.S.) confirmed the articles, and the two authors finally decided whether to include the articles in the discussion.

### 2.5. Outcomes

In this study, we set the prevalence of reporting effect modifications based only on relative scale outcomes and that of inappropriately interpreting the coefficient of interaction terms in nonlinear models on a categorical outcome as the primary outcomes. Regarding outcome scale, we assessed whether researchers reported only a difference in relative scale outcomes as an index of an effect modification, or used the difference in absolute scale outcomes as an index of an effect modification. Regarding the interpretation of interaction terms, we assessed whether researchers interpreted the coefficient of interaction terms in nonlinear models only as an index of an effect modification, or interpreted the coefficient of an index of model fitness and evaluated effect modification using other metrics, such as marginal effects. Moreover, when researchers evaluated an effect modification, but the methodology was not sufficiently described in the main text, supplementary materials, or the study protocol, we considered the study as reporting an “unclear description.” A.S. and one of the coauthors (N.Y., N.S., M.O., or H.S.) assessed the studies independently, and any disagreement was resolved through discussion. When there were multiple articles on a single trial (e.g., salami slicing), we used the number of articles rather than the number of studies.

### 2.6. Data Items

A.S. recorded the funding sources and the number of citations from the Web of Science on 19 June 2022. A.S. confirmed the presence of funding by for-profit organizations through an internet search and specialties (e.g., infectious disease, neurology, and cardiology) and types of intervention (e.g., behavioral intervention, device, medication, and surgery/procedure) through full-text reviews. One of the coauthors (N.Y., N.S., M.O., or H.S.) extracted the following information from the main text, cited protocols, and cited trial registration, and then A.S. confirmed it: methodologies used for evaluating effect modifications; whether any statisticians were included as coauthors; whether the CONSORT statement was cited for reporting; the number of effect modifications evaluated; whether all analyses for effect modifications were pre-specified and described in the main text or the cited protocol; whether multiplicity adjustments (e.g., Bonferroni, Holm or Hochberg methods) were used to evaluate effect modifications; the presence of spin in the abstract or main text based on the results of the effect modifications; and whether statistically significant results of any effect modifications were reported [11]. “Spin” was defined as authors highlighting the result of secondary or explanatory analyses despite a non-significant result for primary outcomes. The definition of “spin” did not include a reporting strategy intended to distract the reader from a non-significant result, unlike in previous literature [12]. A.S. and one of the coauthors (N.Y., N.S., M.O., or H.S.) independently assessed the presence of spin and reached a consensus through discussion.

### 2.7. Statistical Analysis

We summarized the study characteristics as the median and interquartile range for continuous variables and as a percentage for categorical variables. We preliminarily determined whether the following factors were associated with insufficient reporting of effect modifications based only on relative scale outcomes: the number of participants, co-authorship of a statistician, citation of the CONSORT statement, the presence of for-profit organizations, the number of evaluated effect modifications, pre-registration of all analyses for effect modifications, the use of multiplicity adjustment, the presence of statistical significance of any effect modification, and spin. A.S. performed the statistical analyses using R software version 4.0.2 (R Foundation for Statistical Computing, Vienna, Austria).

## 3. Results

### 3.1. Study Characteristics

Figure 1 shows the study selection flowchart. After title and abstract screening and 575 subsequent full-text reviews, we finally included 52 articles describing the analyses of effect modifications based on a categorical outcome (Appendix A). The study characteristics are summarized in Table 1. We included articles from various specialties, such as anesthesiology, cardiovascular surgery, cardiology, critical care, emergency medicine, gastroenterology, general surgery, hematology, immunology, infectious disease, internal medicine, nephrology, neurology, obstetrics and gynecology, oncology, otolaryngology, pediatrics, psychology, pulmonology, rheumatology, and urology.

The methodologies used to evaluate effect modifications varied among the included studies. We considered 4/52 (8%) articles to be reporting unclear descriptions [13,14,15,16]. Regarding non-regression strategies, 5/52 (10%) studies used chi-square tests, F tests, and visual inspection of treatment effects [17,18,19,20,21]. Regarding regression strategies, 2/52 (4%) studies used linear binomial regression to evaluate effect modifications based on absolute scale outcomes [22,23]. Furthermore, 41/52 (79%) studies used a nonlinear regression with interaction terms. Multiplicity adjustments were used in 11 (21%) studies. Regarding the results described in each article, 13 (25%) articles showed at least one statistically significant result of an effect modification, and spin was detected in three (23%) of them.

### 3.2. Primary Outcomes

With respect to outcome scale, in 45/52 (87%) of the included articles and 39/41 (95%) of the articles using nonlinear models, researchers reported effect modification based only on relative scale outcomes to identify patient subgroups. In 4/52 (8%) studies, researchers evaluated effect modifications based on relative and absolute scale outcomes or visual inspection of forest plots [18,20,22,23]. With respect to interpretations of interaction terms, at least 39/41 (95%) articles inappropriately interpreted the coefficient of interaction terms merely as an index of effect modifications. We could not assess the methodology in 2/41 (5%) articles because of unclear descriptions. The individual outcome assessments are summarized in Appendix A.

### 3.3. Exploratory Analyses

The exploratory analyses are summarized in Table 2. Compared with the articles reporting effect modification based not only on relative scale outcomes, citation of CONSORT statement, the statistical significance of any effect modifications, and spin were observed more in articles reporting only relative scale outcomes.

### 3.4. Difference between Protocol and Review

In the protocol, we planned to assess the appropriateness based on an outcome scale used for effect modifications. We changed it to describe the methodologies used in evaluating effect modifications in detail to ensure that readers can easily understand how to improve their methodologies. To avoid mis-specifying the study characteristics, we changed the data extraction and exploratory analyses of the protocol. According to the protocol, A.S. solely decided which articles to include. We modified this protocol to ensure that the decision was made by two authors after discussion. We changed the data source of the number of citations from PubMed to Web of Science and that of specialties from Web of Science to full-text reviews. Although we had planned exploratory statistical tests to evaluate the association between the study characteristics and inappropriateness, we did not conduct these statistical tests owing to the small sample size. Instead, we conducted only descriptive analyses.

## 4. Discussion

Our study had the following two aims: (1) evaluating the prevalence of reporting effect modifications based only on relative scale outcomes and (2) describing the prevalence of inappropriate interpretation of the coefficients of nonlinear models. To the best of our knowledge, this is the first systematic review to comprehensively analyze the prevalence of reporting effect modifications based only on relative scale outcomes and of inappropriately interpreting the coefficient of interaction terms in nonlinear models on categorical outcomes in medical journals with high impact factors. Over 80% of the RCTs reported effect modifications based only on relative scale outcomes and inappropriately interpreted the coefficients of the interaction terms as indices of effect modifications. Our findings convey two important messages: (1) it would be better for researchers to report effect modifications on both absolute and relative scale outcomes, and (2) researchers should not interpret the coefficient of interaction terms in nonlinear regression for categorical outcomes.

Although absolute scale outcomes, such as absolute risk difference, have been recommended for identifying patient subgroups, most RCTs included in this study used nonlinear regression and evaluated effect modifications based only on relative scale outcomes [2,3]. A previous meta-epidemiological study reviewed subgroup analyses of binary outcomes in articles published in *The New England Journal of Medicine* and found that only 40% of these articles reported an absolute risk difference [4]. We summarized the detailed methodologies used for evaluating effect modifications, and most subgroup analyses used interaction terms in nonlinear regression. This was the main reason for reporting effect modifications based only on relative scale outcomes. Although interaction terms can improve model fitness in nonlinear regression, researchers should be aware that nonlinear regression indicates only relative scale outcomes and that absolute scale outcomes should be reported instead [24].

In addition, a substantial proportion of RCTs inappropriately interpreted interaction terms in nonlinear regression analyses to evaluate for effect modifications in the natural metrics (e.g., probability). When researchers use regression for categorical outcomes, it may be difficult to use a single absolute scale, such as the absolute risk difference, to represent the entire population. Although researchers can evaluate interaction terms as absolute scale outcomes using linear probability models, in many cases, these models do not fit the data. They result in predicted probabilities outside zero to one, and predictions outside the logical range are nonsense [25]. The predicted probability of a linear probability model is quite different from the true value when it is close to zero or one. As our study showed, interaction terms in nonlinear regression analyses have been inappropriately evaluated in high-impact-factor medical journals. Instead of interpreting the coefficient of interaction terms, some researchers have proposed recommendations for the appropriate evaluation of effect modifications based on categorical outcomes in nonlinear models [5,8]. One of them used a marginal effect or difference of two predicted outcomes (treatment vs. not) as an absolute scale outcome [7]. Testing the second difference in the two marginal effects across a subgroup could be an appropriate way to evaluate an effect modification. Researchers should appropriately analyze effect modifications, and reviewers should carefully evaluate them.

Our recommendations can be generalized to other medical journals and study designs, such as observational studies and meta-analyses. Our study involved only articles published in high-impact-factor medical journals. However, these journals employ rigorous statistical reviews of submitted manuscripts, and we expect the prevalence of inappropriate evaluations to be higher in other medical journals. Thus, appropriate methodologies for evaluating effect modifications could be a major statistical issue in clinical journals.

Our descriptive and exploratory analyses suggest that there might be room for improvement in the design and reporting of many RCTs. During trial design, researchers should clearly describe the methodology for subgroup analyses in the protocol, especially how they evaluated effect modifications, and should use multiplicity adjustment to avoid alpha errors. When reporting the trial, researchers should avoid overstatement of the secondary or exploratory analyses and cite the CONSORT statement to maintain the quality of reporting. Although spin was detected in only a small number of articles in our study, this may be due to the low number of positive results of subgroup analyses. Our study found that these endeavors have not been fully considered by RCTs.

Our study had several limitations. First, because we could not collect individual patient data, it was not possible to evaluate whether an appropriate evaluation would change the direction of the study results. However, a previous study used three-patient cohort data and showed that different interpretations between absolute and relative scale outcomes are possible [26]. Second, the initial screening of titles and abstracts was conducted by just one author; therefore, selection errors may have occurred. Although we reviewed the full text of 578/686 (84%) articles, the number of articles included in the review may be underestimated. Third, owing to the small number of included studies, we could not identify the factors associated with reporting effect modifications based only on relative scale outcomes. Further meta-epidemiological studies are needed to identify modifiable factors to evaluate effect modifications.

## 5. Conclusions

In our study, the prevalence of reporting effect modification based only on relative scale outcomes, as well as that of inappropriately interpreting the coefficient of interaction terms in nonlinear models, was quite high in high-impact-factor medical journals evaluating categorical outcomes. Researchers should report effect modifications based on both absolute and relative scale outcomes and avoid interpreting the coefficient of interaction terms in nonlinear regressions.

## Figures and Tables

**Figure 1 ijerph-19-15262-f001:**
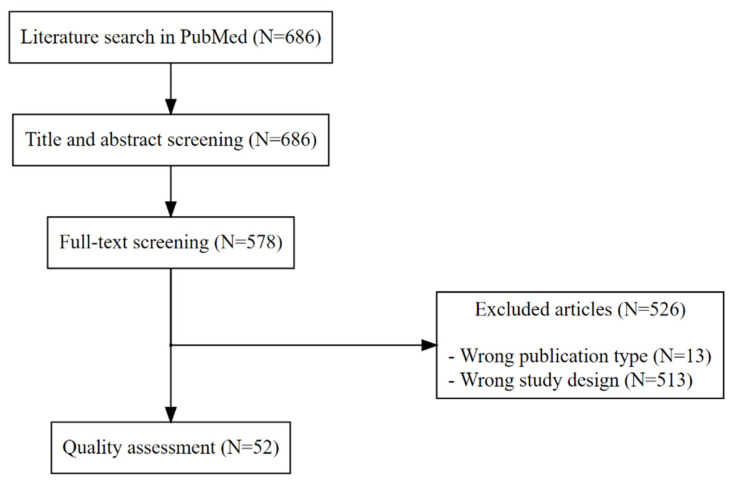
After screening 686 titles and abstracts and subsequently reviewing 578 full articles, we finally included 52 articles. The included and excluded articles are summarized in Appendix A.

**Table 1 ijerph-19-15262-t001:** Study characteristics.

Variable	Result
Total number of included studies (n)	52
Interventions	
Behavioral intervention (n, %)	10 (19)
Device (n, %)	4 (8)
Medication (n, %)	22 (42)
Surgery/procedure (n, %)	16 (31)
Funded by for-profit organizations (n, %)	7 (13)
Number of citations (median, IQR)	9 (5–27)
Co-authorship of a statistician (n, %)	34 (65)
Citation of CONSORT statement (n, %)	17 (33)
Number of effect modifications evaluated (median, IQR)	6 (4–9)
Pre-registration of all analyses for effect modifications (n, %)	41 (79)
Nonlinear regression (n, %)	41 (79)
Generalized linear regression (n, %)	27 (66)
Logistic regression (n, %)	15 (37)
Poisson regression (n, %)	6 (17)
Other (n, %)	6 (17)
Generalized estimating equations (n, %)	2 (5)
Generalized linear mixed-effects model (n, %)	12 (29)
Mixed-effects logistic regression (n, %)	9 (22)
Mixed-effects Poisson regression (n, %)	3 (7)

Note: n, number; IQR, interquartile range.

**Table 2 ijerph-19-15262-t002:** Summary of exploratory analyses.

Variable	Appropriate Evaluation (n = 7)	Inappropriate Evaluation (n = 45)
Co-authorship of a statistician (n, %)	5 (71)	29 (64)
citation of CONSORT statement (n, %)	0 (0)	17 (38)
funded by for-profit organizations (n, %)	1 (14)	6 (14)
the number of effect modifications evaluated (n, IQR)	5 [3 to 8]	6 [4 to 10]
pre-registration of all analyses for effect modifications (n, %)	3 (75)	36 (78)
multiplicity adjustment (n, %)	2 (39)	9 (20)
statistical significance of any effect modifications (n, %)	1 (14)	12 (27)
spin (n, %)	0 (0)	3 (7)

Note: n, number; IQR, interquartile range.

## Data Availability

A.S. had full access to all the data in the study and takes responsibility for the integrity of the data and the accuracy of the data analysis.

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
