# Peer review of "Inappropriate Evaluation of Effect Modifications Based on Categorical Outcomes: A Systematic Review of Randomized Controlled Trials"

_ijerph, 2022, doi:10.3390/ijerph192215262_

Round 1

Reviewer 1 Report

In this systematic review the authors have assessed the quality of reporting effect modification in recently published medical journal articles. 

This work is crucial as it addresses the common error many make when comparing effects or associations across strata of their data. I also appreciate their approach's simplicity, using data from "high-impact" journals in 2021 and their example in the introduction.

It is great that the authors take the "spin"-concept into account. Their definition of spin is somewhat narrower than the publication they refer to (reference 12). I wondered whether they were using the exact definition as ref 12 or if they had modified it. Please be more specific. 

In line 280, they state "In addition, researchers should take caution to avoid spins…." Despite finding spins only in few articles. I think this "spin" statement was more relevant in 2010 than now.

Author Response

Reviewer 1:

#1 In this systematic review the authors have assessed the quality of reporting effect modification in recently published medical journal articles. 

 This work is crucial as it addresses the common error many make when comparing effects or associations across strata of their data. I also appreciate their approach's simplicity, using data from "high-impact" journals in 2021 and their example in the introduction.

Response: We greatly appreciate the reviewer’s endorsement of our manuscript and the reassuring comments.

#2 It is great that the authors take the "spin"-concept into account. Their definition of spin is somewhat narrower than the publication they refer to (reference 12). I wondered whether they were using the exact definition as ref 12 or if they had modified it. Please be more specific. 

Response: Thank you very much for your suggestion. As pointed out, we used a different definition of spin. We have added a more specific description to the methods section in the revised manuscript.

“The definition of spin did not include a reporting strategy intended to distract the reader from a non-significant result, unlike previous literature [12].” (Pages 4-5, Lines 176-178)

#3 In line 280, they state "In addition, researchers should take caution to avoid spins…." Despite finding spins only in few articles. I think this "spin" statement was more relevant in 2010 than now.

Response: We have deleted sentences not supported by our results.

Reviewer 2 Report

This is a useful study showing that – just as is the case in many related fields – medical RCT reports have similar issues when it comes to nonlinear interactions. As someone who has written on this issue I appreciate seeing it being called to attention in this field and also the concrete data from the meta-analysis to show it is indeed a prevalent issue that needs to be addressed.

Overall, I think the contribution is straightforward and useful. I have some specific suggestions for clarity and improving the contribution.

SPECIFIC SUGGESTIONS

Page 2 – I wouldn’t report these hypothetical relative differences as 2% but instead as “2 times” (which is 200%). The key is it is a doubling in the relative scale.

The article doesn’t really explain why the coefficient on the interaction term in a nonlinear model is not a test of interaction in the absolute scale (the introduction in particular is quite sparse). Perhaps it does not need to – but if it does not it would be useful earlier on to state the statistical details are explained elsewhere and provide a recommended cite or two for interested readers. In addition, some brief intuitive explanation of the problem would be useful.

The article is somewhat inconsistent in that there seems to be an implicit that the interaction test in the relative scale is not the correct one. But then the takeaway is to report both the absolute and relative scales. If the relative scales are wrong why report them? In general, I argue for focusing on absolute differences so am sympathetic to the implicit argument. But this is something that should be more directly stated.

One slight disconnect between the opening and the actual studies being assessed is that the opening example focuses on risk differences (absolute) vs risk ratios (relative). However, the coefficients in, e.g. binary logit, are odds ratios (once exponentiated as they usually are for interpretation). So not only are the relative effects but they also involve a nonlinear transformation of the effects into the odds scale. This is one of the reasons the coefficients can differ so dramatically from the conclusions in the absolute (e.g. predicted probability) metric.

I am not an expert in meta-analysis but the inclusion criteria and study selection seems well done.

The results are presented in a straightforward and understandable way.

Page 7: “They produce predicted probabilities outside zero to one, and the models are 256 most significant when most probabilities are close to zero or one [25].” – This sentence needs rewritten for clarity. Avoiding the word “significant” when not referencing statistical significance would help.

Author Response

Reviewer 2:

#1 This is a useful study showing that – just as is the case in many related fields – medical RCT reports have similar issues when it comes to nonlinear interactions. As someone who has written on this issue I appreciate seeing it being called to attention in this field and also the concrete data from the meta-analysis to show it is indeed a prevalent issue that needs to be addressed. Overall, I think the contribution is straightforward and useful. I have some specific suggestions for clarity and improving the contribution.

Response: We are grateful for your thoughtful comments and suggestions.

SPECIFIC SUGGESTIONS

#2 Page 2 – I wouldn’t report these hypothetical relative differences as 2% but instead as “2 times” (which is 200%). The key is it is a doubling in the relative scale.

Response: We are sorry for the confusing description. We have modified the relevant instances according to your suggestion.

“16%/8% = 2 times2” and “8%/4% = 2 times” (Page 2, Line 65)

#3 The article doesn’t really explain why the coefficient on the interaction term in a nonlinear model is not a test of interaction in the absolute scale (the introduction in particular is quite sparse). Perhaps it does not need to – but if it does not it would be useful earlier on to state the statistical details are explained elsewhere and provide a recommended cite or two for interested readers. In addition, some brief intuitive explanation of the problem would be useful.

Response: We appreciate your suggestions. We have added more explanations on why the coefficients of the interaction terms in a nonlinear model do not indicate effect modification as follows:

“The same logic applies to regression. In linear regression analyses, the coefficients indicate an absolute scale, while in nonlinear regression analyses, these indicate a relative scale. For example, a coefficient of a logistic regression indicates the change of the outcome on a log-odds ratio scale relative to a one-unit change in the corresponding covariate. It is caused by the link function (e.g., logit for logistic regression and log for Poisson regression). Furthermore, because of the link function, nonlinear regression analyses show an inherent interaction, i.e., treatment effects are constant on a relative scale (e.g., the log-odds scale for logistic regression), whereas the probability of the outcome changes depending on other variables without interaction terms [5, 6]. For instance, as illustrated in Supplementary Figure 1, a logit curve moves upwards when the intercept only increases. The change in probability is small at both extreme dependent values while it is large at the middle (i.e., compression). This phenomenon is of an inherently interactive nature because the difference between the two logistic curves is contingent on the value of an independent variable. It explains the interactions between one independent variable and another independent variable without any interaction terms. In other words, just including a different independent variable without the interaction term changes the residual variation of the underlying model and moves a logit curve similarly. This inherent interactive nature was summarized by Mize, et al [7]. Thus, the coefficients of the interaction terms do not necessarily indicate any effect modification” (Pages 2-3, Line 70-88)

Supplement Figure 1. Inherent interactive nature of logistic regression

These two curves represent the logistic curves. When the intercept only changes, the black curve moves toward the red curve. The difference in the two logistic curves is not constant and is contingent on an independent variable. This phenomenon explains the inherently interactive nature of a logistic regression model. It is the same when using other non-linear models such as Poisson regression and the Probit model because of the link function. 

# 4 The article is somewhat inconsistent in that there seems to be an implicit that the interaction test in the relative scale is not the correct one. But then the takeaway is to report both the absolute and relative scales. If the relative scales are wrong why report them? In general, I argue for focusing on absolute differences so am sympathetic to the implicit argument. But this is something that should be more directly stated.

Response: As the reviewer mentioned, our sentences were confusing. Our key messages were the following: (1) it would be better for researchers to report effect modifications on both absolute and relative scale outcomes, and (2) researchers should not interpret the coefficients of interaction terms in nonlinear regression for categorical outcomes. We revised the manuscript for bringing clarity on these key points.

Our study had the following two aims: (1) evaluating the prevalence of reporting effect modifications based only on relative scale outcomes and (2) describing the prevalence of inappropriate interpretation of the coefficients of nonlinear models. (Page 6, Lines 244-246)

Our findings convey two important messages– (1) it would be better for researchers to report effect modifications on both absolute and relative scale outcomes, and (2) researchers should not interpret the coefficient of interaction terms in nonlinear regression for categorical outcomes.

(Page 7, Lines 252-255)

#5 One slight disconnect between the opening and the actual studies being assessed is that the opening example focuses on risk differences (absolute) vs risk ratios (relative). However, the coefficients in, e.g. binary logit, are odds ratios (once exponentiated as they usually are for interpretation). So not only are the relative effects but they also involve a nonlinear transformation of the effects into the odds scale. This is one of the reasons the coefficients can differ so dramatically from the conclusions in the absolute (e.g. predicted probability) metric.

Response: Thank you very much for the clarification. We have added detailed information in the manuscript as described above.

#6 I am not an expert in meta-analysis but the inclusion criteria and study selection seems well done. The results are presented in a straightforward and understandable way.

Response: We appreciate the reviewer’s thoughtful comment. 

#7 Page 7: “They produce predicted probabilities outside zero to one, and the models are 256 most significant when most probabilities are close to zero or one [25].” – This sentence needs rewritten for clarity. Avoiding the word “significant” when not referencing statistical significance would help.

Response: We apologize for the unclear sentences. We have clarified them in the revised manuscript.

They result in predicted probabilities outside zero to one, and predictions outside the logical range are nonsense [25]. The predicted probability of a linear probability model is quite different from the true value when it is close to zero or one. (Page 7, Lines 274-276)